# OpenReview forum: "Misusing Tools in Large Language Models With Visual Adversarial Examples"
_ICLR.cc/2024/Conference — Submitted to ICLR 2024_

### Official Review · Reviewer_cgCA · 2023-10-20

**Soundness:** 4 excellent
**Presentation:** 4 excellent
**Contribution:** 3 good
**Rating:** 6
**Confidence:** 5

**Summary:**

This paper studies prompt injection against large language models (LLMs) through the visual modality, i.e., crafting (universal) adversarial image examples to cause the LLM to invoke tools following real-world syntax. In particular, the authors highlight the consideration of attack stealthiness in terms of both perturbation imperceptibility and response utility. Five variants of attacks with varied attack difficulty are considered.

**Strengths:**

- Prompt injection against LLMs is a promising direction, especially through the relatively new channel of visual modality.
- The paper is very well written, including sufficient example visualizations and clearly described technical details.
- Experiments are extensive and insightful, which include human studies and ablation studies of important hyperparameters.

**Weaknesses:**

- The best result is reported among three trials based on the argument that “attackers will always choose the best-performing adversarial image.” However, the reviewer thinks this may not be reasonable because the attacker is not the user and so cannot control how many times they would repeat the attack. The authors should explain why they think it is feasible to stick to this setting. This is important given the fact that significant randomness during adversarial image training is observed.

- The authors motivate the design of separate losses for response utility and tool invocation (i.e., Equation 2) based on the argument “In real-world conversational systems...we reduce the contribution of the loss term...” First of all, there are comparisons validating the superiority of using such separate losses to the integrated loss (i.e., Equation 1). More specifically, the ablation studies clearly show using a large $\lambda$, i.e. 1, works better, which conflicts with the argument about “reducing the contribution...”.

- It is said that “the l2 norm is computed with regard to each color channel separately”. However, to the best knowledge of the reviewer, in the literature of adversarial examples, it is indeed computed not separately. Could the authors explain why they chose this unusual setting?

- Considering that the threat model follows the typical prompt injection, it seems unnecessary to highlight its differences from “jailbreaking”. Therefore, the authors are encouraged to tune down some related claims.

- The fact that only three images are tested should be mentioned in the main text rather than only in the appendix.

**Questions:**

See the above weaknesses.

---

> ### Author Response · Authors · 2023-11-17
>
> Thank you for your valuable feedback! Here are our responses to your questions and suggestions.
>
> > The best result is reported among three trials based on the argument that “attackers will always choose the best-performing adversarial image.” However, the reviewer thinks this may not be reasonable because the attacker is not the user and so cannot control how many times they would repeat the attack. The authors should explain why they think it is feasible to stick to this setting. This is important given the fact that significant randomness during adversarial image training is observed.
>
> It’s 100% true that the attacker has no control over the user, and it’s the exact reason  why the attacker wants the image that is the best performing such that it is expected to affect the most victims.
>
> > The authors motivate the design of separate losses for response utility and tool invocation (i.e., Equation 2) based on the argument “In real-world conversational systems...we reduce the contribution of the loss term...” First of all, there are comparisons validating the superiority of using such separate losses to the integrated loss (i.e., Equation 1). More specifically, the ablation studies clearly show using a large lambda, i.e. 1, works better, which conflicts with the argument about “reducing the contribution...”.
>
> We will update the text to “attempt to reduce the contribution” as we designed this loss based on intuitions and later experimented with them. During our experiments, we found that lambda_i provides a good tradeoff between imperceptibility and goal achieving (Table 7), while the effect of lambda_r is more ambivalent (Table 6) — while the response utility drops, the goal achieving likelihood for the unrelated set increases during (lambda_r 1.0 -> 0.1) but further decreases in the range of (lambda_r 0.1 -> 0), while mostly stays the same for the related set. Based on the exact values, we believe an attack that has higher response utility is more favorable to the attacker, therefore in the main experiments, we went with a lambda_r = 1.0. Thanks for pointing this out.
>
> > It is said that “the l2 norm is computed with regard to each color channel separately”. However, to the best knowledge of the reviewer, in the literature of adversarial examples, it is indeed computed not separately. Could the authors explain why they chose this unusual setting?
>
> It is actually a historical issue where our initial attempts adopt different normalization coefficients separately for each channel in the preprocessing stage, making us also calculate the l2 norm separately. Given that using both formats can achieve the goal of minimizing the difference between the adversarial image and the original one, we kept the previous way of computing and reported it in the paper for reproducibility.
>
> > Considering that the threat model follows the typical prompt injection, it seems unnecessary to highlight its differences from “jailbreaking”. Therefore, the authors are encouraged to tune down some related claims.
>
> Thanks for your suggestion. We have shortened the discussion about the jailbreaking issue in section 2 to make room for other revisions.
>
>
> >  The fact that only three images are tested should be mentioned in the main text rather than only in the appendix.
>
> Thanks for pointing this out. We will make this more explicit in the main text. We picked the three images from different sources (we described in the appendix that one the logo of the visual language model we are using, one generated through stable diffusion, and one from online sources), hoping that the present a good coverage over image types. We happily triple the number of images and provide six more examples in Table 9 (Appendix A.9), including two from the Imagenet dataset, and two more each coming from model generations and online sources. We couldn’t conduct a larger experiment with more images due to time constraints but we hope that this provides a good indicator that the attack method would work generally on many images.

---

> > ### Comment · Reviewer_cgCA · 2023-11-20
> > **thanks for your response**
> >
> > Q1. I still think it is improper to report the best attack performance rather than the average performance (with std.). The authors may describe more details about when the attack happened. As the authors stated, if the attacker can control the image completely, it should happen before the user touches that image.
> >
> > Q2. I do not quite understand the response, does the author mean "They try to do that but finally fail?" If so, I don't think simply tweaking the wording is enough. Since the loss term does not work as expected, it should be not considered at all.
> >
> > For the other concerns, the authors have promised to address them in the final version.

---

> > > ### Author Response · Authors · 2023-11-21
> > > **Thanks for the follow up!**
> > >
> > > Thanks, please check our responses below.
> > >
> > > > I still think it is improper to report the best attack performance rather than the average performance (with std.). The authors may describe more details about when the attack happened. As the authors stated, if the attacker can control the image completely, it should happen before the user touches that image.
> > >
> > > There seems to be a misunderstanding here. The scenario we are imagining is: the attacker would choose the image that is most successful among several trials they attempted themselves with the attack. Therefore, our experiments simulate this setting by reporting the best performance among three trials. The reviewer might be worried that we are reporting the best performance of each individual instance (across three trials) and then reporting the average across the dataset. We clarify that this is not the case, as we are obtaining the average performance of all instances for each adversarial image, and then reporting the best one. This mostly rules out completely failing attack trials, which the attacker can easily realize by trying some inputs. In our early experiments, we also splitted 20 instances from the test set as a development set of model selection, which almost always led to the same selected model, hence we omitted such a detour. On a side note,  the transferability to out-domain instances should also justify the robustness of the (successful) adversarial images.
> > >
> > > > I do not quite understand the response, does the author mean "They try to do that but finally fail?" If so, I don't think simply tweaking the wording is enough. Since the loss term does not work as expected, it should be not considered at all.
> > >
> > > We like to clarify that the hyperparameter is not “failing,” as it does relate with response utility. It doesn’t seem to be always working against the attack objective, as one would expect for a tradeoff hyperparameter. In this sense, we do agree that this hyperparameter could be omitted (or better, put in a separate section called unsuccessful attempts) had the paper’s main focus only be about a new attack technique. However, our paper’s major focus has been on emphasizing such an attack scenario (and the possibility of such attack), therefore the understanding of such a hyperparameter that controls response utility is worth studying and highlighting. That’s why we believe discussing it in the main text would provide better information to the reader.

---

> > > > ### Comment · Reviewer_cgCA · 2023-11-21
> > > > **Thanks for the further reply.**
> > > >
> > > > My previous concerns are addressed. I still lean toward accepting the paper considering the new threat model and comprehensive explorations. However, I do agree with the reviewers that testing the attack against a real-world VLM system is needed, especially considering that the attack focuses on application-level threats.

---

### Official Review · Reviewer_La6s · 2023-10-29

**Soundness:** 2 fair
**Presentation:** 3 good
**Contribution:** 2 fair
**Rating:** 3
**Confidence:** 4

**Summary:**

This paper presents a method to cause malicious tool usage of LLM by using adversarial examples. The idea is to use a loss that includes both the response utility and malicious behavior for training

**Strengths:**

1. The paper studies a new and timely security issue of LLM.
2. The proposed method achieves better stealthiness over prior works.

**Weaknesses:**

1. The proposed method seems to be straightforward, which is essentially the way of injecting a backdoor. It is unclear how this method can help generalize the trigger to other prompts.

2. Since the goal is to trigger the misusage of tools, why limiting the adversarial perturbation and enhancing the generalizability are important? The adversary only needs one prompt to trigger the malicious usage of the tools.

3. It is unclear what are the implications of these 5 attack objectives. Does the selection of these attack objectives have an impact on the attack performance? What will happen if more attack objectives are included?

4. The evaluation is unsatisfactory. L_p norm is not provided. Also, it is important to show the chances that the malicious tool is triggered when the adversarial example and prompt pair are not present, or only one of these two is present.

**Questions:**

See Weaknesses.

**Details Of Ethics Concerns:**

The paper presents an attack against LLMs that are integrated with third-party tools. However, ethics concerns are not discussed.

---

> ### Author Response · Authors · 2023-11-17
>
> > The proposed method seems to be straightforward, which is essentially the way of injecting a backdoor. It is unclear how this method can help generalize the trigger to other prompts.
>
> We refer the reviewer to the response for Review tv67 above for reasons as to why our contribution matters. To briefly summarize that response, our contribution is in the computer security space – we show a new threat model for multi-modal LLMs. The fact that an attack is straightforward is a big advantage because attackers always choose the path of least resistance. Furthermore, given that there is 10 years of work on improving visual adversarial example attacks, all of that is now applicable to multi-modal LLMs – this is a formidable arsenal for attackers targeting LLMs. We view our work as opening the flood-gates of attacks on LLMs. It also serves as strong motivation to start investing in research to defend against these attacks.
>
> Table 4 and 5 establishes that our method generalizes to multiple types of prompts and function invocation syntaxes.
> Beyond that, with regards to the method itself, being able to generalize our attack to various or even arbitrary text prompts provided by the victim user with a decent success rate (on desired tool invocation), while maintaining the response and image stealthiness is a non-trivial task itself and has never been accomplished by any prior work in this domain. Additionally, an attack that is not prohibitively complicated in its nature means it’s more dangerous from a security perspective since it’s more easily deployable and more likely to be used by real world attackers.
>
> > Since the goal is to trigger the misusage of tools, why limiting the adversarial perturbation and enhancing the generalizability are important? The adversary only needs one prompt to trigger the malicious usage of the tools.
>
>
> Attackers ultimately want the attack to be spread wide and long enough to affect as many victims as possible. The image and response stealthiness are critical in this sense because victims will otherwise easily spot the abnormality of the image or the conversation and may even report this to the model provider.
>
> > It is unclear what are the implications of these 5 attack objectives. Does the selection of these attack objectives have an impact on the attack performance? What will happen if more attack objectives are included?
>
> We chose these five attack objectives to simulate the real-world behavior as realistically as possible by following invocation format from real world systems (see Section 3.1). In fact, the string <function.delete_email which="all"> is the function call format specified in an [older version](https://github.com/microsoft/semantic-kernel/blob/334b22a78460a46b91391c1b41f79e23d55338c2/dotnet/src/Extensions/Planning.SequentialPlanner/skprompt.txt) of Microsoft Semantic Kernel. And the Expedia invocation json is an exact copy of the one generated by ChatGPT (and json is the [standard format](https://platform.openai.com/docs/guides/function-calling) for any plugin calls in ChatGPT). We believe it’s a fair claim that the textual format attack objectives we’ve tested does reflect practical scenarios and demonstrates the potential of our attack if they could be applied on real-world systems. We have revised the paper to clarify.
> Also, these five attack objectives represent increasing levels of complexity and difficulty to generate the desired attack and serve as examples to understand whether it is still possible and how easy/difficult it is when the target texts to be generated by LLMs are non natural and complicated.

---

> > ### Comment · Reviewer_La6s · 2023-11-23
> > **Thanks for the response!**
> >
> > I appreciate the authors' response. However, the threat model and the setting of the problem are still not very well motivated. The function invocation can be easily addressed by a safeguard after the LLM, if we consider an adaptive defender, which we should do in practice. Besides, in the proposed framework, function invocation can be replaced by any other payloads. It is still not clear to me why enhancing the generalizability is important. Similarly, if we consider an adaptive defender, "Attackers ultimately want the attack to be spread wide and long enough to affect as many victims as possible" may not be a valid assumption.

---

> ### Author Response · Authors · 2023-11-17
>
> >  The evaluation is unsatisfactory. L_p norm is not provided. Also, it is important to show the chances that the malicious tool is triggered when the adversarial example and prompt pair are not present, or only one of these two is present.
>
> We additionally provide the L_2 norm and L_inf norm between the adversarial image and its original in Table 4. (Note that Table 5 corresponds to the same adversarial images thus the L_p norms would be the same if provided). Additionally, in appendix Table 7, we provide the L_p norms to understand the quantities when the SSIM score is low; in appendix figure 5, a plot of SSIM values and L_2 norm for experiments in the paper is provided, which shows that SSIM is almost (inverse) proportional to the L_2 norms (as expected).
>
> Regarding the other scenarios of the attack: we are interested in the scenario when the attacker specified tool invocation should appear when the adversarial example is __present__ with an __arbitrary__ prompt provided along with it. When there is no adversarial image provided but only a text prompt, the LLM should respond to the prompt as usual. In the rare case when the user provides no text prompt provided along with the adversarial image, the LLM we tested would respond as if there is a prompt asking “describe this image”, and for several of our trained adversarial images, also generate the tool invocation.

---

### Official Review · Reviewer_NtXf · 2023-10-31

**Soundness:** 3 good
**Presentation:** 3 good
**Contribution:** 2 fair
**Rating:** 5
**Confidence:** 4

**Summary:**

This paper proposes using visual adversarial examples to attack vision-integrated LLMs. Particularly, this paper focuses on attacking the LLMs to affect the confidentiality and integrity of users' resources connected to the LLMs. The proposed attack is stealthy (as visual adversarial examples look similar to normal images), and generalizable (the adversarial examples can trigger the targeted generation when paired with different text prompts). And it is shown that the attack can successfully trigger LLMs to output the tools-abusing texts.

**Strengths:**

1. The big picture of the paper is sound. Indeed, as LLMs are integrated into applications, critical resources may be controlled by the models. Then, attacks on the models can induce broad implications beyond just the misalignment moral values. The threat model and the real-world risk analysis in this paper are quite insightful.

2. The approach is simple and effective.

3. The authors make efforts to collect evaluation datasets as well as comprehensive human evaluation.

**Weaknesses:**

1. **Only a single model LLaMA Adapter is tested.** This makes the scope of the evaluation look somewhat narrow. I suggest the authors also consider other VLMs like Minigpt-4 [1], Instruct-Blip [2], and LLaVA [3]. This can make the evaluation more convincing.

2. **Lack of case studies on real LLM-integrated applications.**  The paper mentioned that LangChain and Guidance facilitate the development of such integrations. But, the paper did not provide a single instance of this to illustrate the practical risks of the proposed attack. In the whole paper, what the attack did was just to induce the generation of something like <function.delete_email which="all"> in a purely textual form, which is essentially nothing different from previous NLP attacks that induced certain targeted generations. In my opinion, the novelty of this paper only comes from the illustration of the "practical risks" of such attacks --- because the resources controlled by LLMs can now also be missed to induce broader harms. However the paper did not provide a real example of this. The whole evaluation is still in a purely textual form, judging whether things like <function.delete_email which="all"> are generated... In practice, the LLMs integrated systems may be more complicated than this conceptual form. The paper did not go deep into this.

3. **Inaccurate Literature Review.** There is a factual error in the literature review. Qi et. al. [4] did not show transferability to closed-source models. On the other hand, Carlini et. al. [5] along with Qi et. al. [4] are earlier works showing the usage of visual adversarial examples to hack VLM, which may also need to be noted.



[1] Zhu, D., Chen, J., Shen, X., Li, X. and Elhoseiny, M., 2023. Minigpt-4: Enhancing vision-language understanding with advanced large language models. arXiv preprint arXiv:2304.10592.

[2] Dai, W., Li, J., Li, D., Tiong, A.M.H., Zhao, J., Wang, W., Li, B., Fung, P. and Hoi, S., 2023. InstructBLIP: Towards General-purpose Vision-Language Models with Instruction Tuning. arXiv preprint arXiv:2305.06500.

[3] Liu, H., Li, C., Wu, Q. and Lee, Y.J., 2023. Visual instruction tuning. arXiv preprint arXiv:2304.08485.

[4] Qi, X., Huang, K., Panda, A., Wang, M. and Mittal, P., 2023, August. Visual adversarial examples jailbreak aligned large language models. In The Second Workshop on New Frontiers in Adversarial Machine Learning.

[5] Carlini, N., Nasr, M., Choquette-Choo, C.A., Jagielski, M., Gao, I., Awadalla, A., Koh, P.W., Ippolito, D., Lee, K., Tramer, F. and Schmidt, L., 2023. Are aligned neural networks adversarially aligned?. arXiv preprint arXiv:2306.15447.

**Questions:**

Can the attack be directly applied to realistic LLM-integrated applications in the wild? Say, other prompt injection attacks such as [1,2] do show realistic instances.


[1] Greshake, K., Abdelnabi, S., Mishra, S., Endres, C., Holz, T. and Fritz, M., 2023. Not what you’ve signed up for: Compromising Real-World LLM-Integrated Applications with Indirect Prompt Injection. arXiv preprint arXiv:2302.12173.

[2] Liu, T., Deng, Z., Meng, G., Li, Y. and Chen, K., 2023. Demystifying RCE Vulnerabilities in LLM-Integrated Apps. arXiv preprint arXiv:2309.02926.

---

> ### Author Response · Authors · 2023-11-17
>
> Thank you for your valuable feedback! Here are our responses to your questions and suggestions.
>
> >Only a single model LLaMA Adapter is tested. This makes the scope of the evaluation look somewhat narrow. I suggest the authors also consider other VLMs like Minigpt-4 [1], Instruct-Blip [2], and LLaVA [3]. This can make the evaluation more convincing.
>
> We acknowledge this and we are currently working on experiments on other multimodal models. The result will be presented in the updated version soon.
>
> > Lack of case studies on real LLM-integrated applications. The paper mentioned that LangChain and Guidance facilitate the development of such integrations. But, the paper did not provide a single instance of this to illustrate the practical risks of the proposed attack. In the whole paper, what the attack did was just to induce the generation of something like <function.delete_email which="all"> in a purely textual form, which is essentially nothing different from previous NLP attacks that induced certain targeted generations. In my opinion, the novelty of this paper only comes from the illustration of the "practical risks" of such attacks --- because the resources controlled by LLMs can now also be missed to induce broader harms. However the paper did not provide a real example of this. The whole evaluation is still in a purely textual form, judging whether things like <function.delete_email which="all"> are generated... In practice, the LLMs integrated systems may be more complicated than this conceptual form. The paper did not go deep into this.
>
> We tried to simulate the real-world behavior as realistically as possible by following invocation format from real world systems (see Section 3.1). In fact, the string <function.delete_email which="all"> is the function call format specified in an [older version](https://github.com/microsoft/semantic-kernel/blob/334b22a78460a46b91391c1b41f79e23d55338c2/dotnet/src/Extensions/Planning.SequentialPlanner/skprompt.txt) of Microsoft Semantic Kernel. And the Expedia invocation json is an exact copy of the one generated by ChatGPT (and json is the [standard format](https://platform.openai.com/docs/guides/function-calling) for any plugin calls in ChatGPT). We believe it’s a fair claim that the textual format attack objectives we’ve tested does reflect practical scenarios and demonstrates the potential of our attack if they could be applied on real-world systems. We have revised the paper to clarify.
> Also, compared to previous work’s targets, a crucial difference of our attack goal is that these tool invocation texts are non natural. We believe it is important to understand whether it is still possible and how easy/difficult it is when the target texts are non natural. Our evaluation provides some answers to this question, with various length of the target text as another dimension.
>
> > Inaccurate Literature Review. There is a factual error in the literature review. Qi et. al. [4] did not show transferability to closed-source models. On the other hand, Carlini et. al. [5] along with Qi et. al. [4] are earlier works showing the usage of visual adversarial examples to hack VLM, which may also need to be noted.
>
> Thank you for pointing these out!  We have corrected these errors in the revised version.

---

> > ### Comment · Reviewer_NtXf · 2023-11-19
> > **Response to authors**
> >
> > Thanks for the revision. However, one concern still remains that --- essentially, the whole work is merely to trigger certain targeted text outputs. There are no realistic case studies to show the misuse of tools in practical setups. This makes the contribution of this work unclear, as also pointed out by Reviewer tv67.

---

> > > ### Author Response · Authors · 2023-11-21
> > > **Thanks for the follow up!**
> > >
> > > Thanks, please check our response below.
> > >
> > > > Thanks for the revision. However, one concern still remains that --- essentially, the whole work is merely to trigger certain targeted text outputs. There are no realistic case studies to show the misuse of tools in practical setups. This makes the contribution of this work unclear, as also pointed out by Reviewer tv67.
> > >
> > > We do not have case studies on commercial LLMs with plugins/tools at the time of submission. This work envisions that such attacks could happen for the commercial LLMs if they are integrated with tool use and multimodal support. This envision is probably being more realistic as the recent release of the GPTs series [1], which matches our expectation that LLMs will be integrated with tool use and image understanding [2]. Our attack is a white-box attack, and since the mostly used LLMs (ChatGPT & Bard) do not have their weights open-sourced, we study open-sourced LLMs following real tool use invocation formats (e.g. OpenAI & Microsoft Guidance, see Section 3.1). Our work, therefore, is a case study since the specific strings we're causing the model to generate are real API calls and are harmful API calls. It indicates that if an attacker is able to access the weights of a LLM product with multimodality support and tool invocations, there is a high risk that such an attack can be executed. This work should raise awareness to LLM builders and users, because of the highly malicious threats that could be carried out from the attack (email manipulation, etc).
> > >
> > > [1] https://openai.com/blog/introducing-gpts
> > >
> > > [2] https://help.openai.com/en/articles/8400551-image-inputs-for-chatgpt-faq

---

### Official Review · Reviewer_tv67 · 2023-11-05

**Soundness:** 2 fair
**Presentation:** 3 good
**Contribution:** 2 fair
**Rating:** 3
**Confidence:** 3

**Summary:**

This paper proposes an attack on multi-modality models that are capable of using tools. The main idea of this paper is to perturb the image side so that the text side can generate malicious instructions that may impact downstream tools. The paper demonstrates that their attack is effective, stealthy, and generalizable.

**Strengths:**

- In general, this paper is well-structured and easy to follow.
- I believe the problem this paper addresses is highly significant. It focuses on understanding how to attack systems using LLMs in real-world scenarios, which presents new challenges when viewed from a systemic perspective.
- The experimental results demonstrate that malicious instructions can be generated by perturbing the input image.

**Weaknesses:**

- This paper assumes that interaction with the tools occurs through an instruction line, followed by normal question answering, as illustrated in Figure 1. Is this setting realistic? What does a real system look like, and how do these VLMs interact with downstream tools like email? Please provide an illustration of why the task in Figure 1 is realistic.
- This paper lacks technical contributions and depth. The technical contribution of this paper is to generate perturbations on the image side that can prompt the language model to output specific words. However, this paper does not discuss the technical challenges associated with this attack scenario. Instead, they employ a simple gradient-based technique widely adopted in adversarial example attacks. They also do not provide an in-depth analysis of the drawbacks of this technique. For example, is this attack easily bypassed? What is the robustness of this attack?
- Additionally, the authors claim that the attack is stealthy because it does not alter the semantic meanings of the output answers. They support this claim with the evidence that the answers under attack are 10% less natural compared to the original ones. My question is, why is a 10% difference considered a small one?
- If the current setting is realistic, I suggest showing the effectiveness of attacking the real-world system.

**Questions:**

To conclude, I think the problem this paper would like to address is attractive. However, it is not clear why the current setting is realistic. Also, the technique contributions and discussion depth hinder its acceptance. I believe these questions are hard to be properly addressed during the rebuttal period.

---

> ### Author Response · Authors · 2023-11-17
>
> Thank you for your feedback. We believe there might be a miscommunication  of our motivation and contributions. We attempt to better clarify as follows.
>
> > This paper assumes that interaction with the tools occurs through an instruction line, followed by normal question answering, as illustrated in Figure 1. Is this setting realistic? What does a real system look like, and how do these VLMs interact with downstream tools like email? Please provide an illustration of why the task in Figure 1 is realistic.
>
> Commercial LLMs like ChatGPT and Google Bard support visual images as an input modality by default. Their interaction method assumes an image being uploaded with a prompt alongside, exactly like what we have described in Figure 1. They are also integrating a large number of extensions/plugins which include ones necessary for our attack objective such as email connectors , [google workspace](https://blog.google/products/bard/google-bard-new-features-update-sept-2023/), Expedia, etc. A version of our attack on these models would successfully invoke the plugins/extensions integrated by the victim user. We note that the attacker-desired invocation string can be changed to invoke any other plugin/extension than the 5 attack objectives we have shown in Table 1). **Therefore, our proposed attack scenario is highly realistic even at this moment.**  The trend of more third-party customized versions of GPT ([GPTs](https://openai.com/blog/introducing-gpts)) that will incorporate more powerful tools to LLMs in custom domains will only make our attack more applicable. We have revised the paper to clarify this.
>
> > This paper lacks technical contributions and depth. The technical contribution of this paper is to generate perturbations on the image side that can prompt the language model to output specific words. However, this paper does not discuss the technical challenges associated with this attack scenario. Instead, they employ a simple gradient-based technique widely adopted in adversarial example attacks. They also do not provide an in-depth analysis of the drawbacks of this technique. For example, is this attack easily bypassed? What is the robustness of this attack?
>
> From a computer security perspective, we contribute a new threat model — an attacker can create stealthy images that result in tool misuse. This shows that the past 10 years of attack research on vision adversarial examples can be applied to multi-modal LLMs. This highlights a critical new class of vulnerabilities in LLMs. Furthermore, given that there isn’t an airtight defense for visual adversarial examples, it implies that multimodal LLMs are fundamentally vulnerable to an attack vector with no known good defenses. Furthermore, attackers always prefer the path of least resistance. The simpler an attack technique is, the more likely that attack will actually occur. Thus, we view the simplicity of the attack technique in the paper as an advantage. Finally, from an ML perspective, we observe that a challenge was to balance three competing objectives: (1) getting the LLM to produce a very specific function invocation syntax; (2) Maintaining a natural-semantics conversation with the user; and (3) Keeping the image modifications stealthy.

---

> > ### Comment · Reviewer_tv67 · 2023-11-17
> > **Thanks for the reply.**
> >
> > 1. For the realistic issue, I don't see why the output format, i.e., natural question answer plus an invocation comment, is realistic for commercial VLMs.
> > 2. For the contribution, I think there are several works that already prove adversarial attacks on the image side can let VLM models output attacker-desired texts (please search for them in google scholar). I don't see a fundamental difference between this paper and these methods, except the desired text in this context is the invocation strings.

---

> > > ### Author Response · Authors · 2023-11-21
> > > **Thanks for the follow up!**
> > >
> > > Thanks, hope our responses below clarify your questions!
> > >
> > > > For the realistic issue, I don't see why the output format, i.e., natural question answer plus an invocation comment, is realistic for commercial VLMs.
> > >
> > > LLMs invoke tools by generating certain tool syntax (invocation comment). We will first introduce an example of tool use through a toy example, then cover why commercial LLMs follow such a syntax, finally discuss the natural question answer before the invocation.
> > >
> > > The example we believe is easy to understand tool using formats is the gsm8k dataset [1] which contains a calculator tool use syntax. Part of the data instance would look like
> > > - The booth made \$50 \times 3 = \$<<50*3=150>>150
> > >
> > > The LLM is expected to generate from left to right the tokens <<50*3=, when here, an external checker uses string matching/regex and realizes that a calculator tool should be called. It invokes the calculator, and places 150>> back to the LLM such that it continues its generation.
> > > Current commercial LLM follow this way to tool use as well. In Section 3.1 we gave references to OpenAI and Microsoft Guidance (semantic kernel) that uses certain tool use syntax (specifically, json and html). Other agent frameworks, such as AutoGPT/Langchain all have their own tool use syntax, nevertheless, are all based on some tool syntax.
> > >
> > > Finally, the natural question answer disguises the unintended tool invocation from the user as the user wouldn’t realize that an unintended tool invocation has happened if the response looks normal.
> > >
> > > As a minor note, we do not differentiate VLMs with LLMs in this response as it is becoming evident that the commercial language models (ChatGPT, BARD, Claude) can take in images or files in addition to text.
> > >
> > > > For the contribution, I think there are several works that already prove adversarial attacks on the image side can let VLM models output attacker-desired texts (please search for them in google scholar). I don't see a fundamental difference between this paper and these methods, except the desired text in this context is the invocation strings.
> > >
> > > We’d like to restate our contribution compared to prior work  (see final paragraph in Section 1 and Section 2, Threat Model):
> > >
> > > From the threat model side:
> > >
> > > (1) we introduce attacking LLMs to generate harmful tool syntax, which we envision to be more and more realistic as LLMs would be integrated with more tool uses. In hindsight, this envision is probably being more realistic as the recent release of the GPTs series [2].
> > >
> > > (2) Different from prior work, we consider triggering tool syntax and generalizing to different user prompts, with the goal of a disguised attack (small image difference and good response utility). Triggering tools provides a high security risk, such as email manipulation.
> > >
> > > Since the mostly used LLMs (ChatGPT & Bard) do not have their weights open-sourced, we use open-sourced LLMs following real tool use invocation formats (e.g. OpenAI & Microsoft Guidance, see Section 3.1) as a case study. We show that with a simple gradient-based attack:
> > >
> > > (1) Generalizability, Stealthiness and Tool abusing can be simultaneously achieved.
> > >
> > > (2) The existence of such a simple attack further raises awareness from a security perspective, furthermore, increases the possibility that the attack could be executed on commercial LLMs had the attacker have access to its weights.
> > >
> > > [1] https://github.com/openai/grade-school-math
> > >
> > > [2] https://openai.com/blog/introducing-gpts

---

> ### Author Response · Authors · 2023-11-17
>
> > Additionally, the authors claim that the attack is stealthy because it does not alter the semantic meanings of the output answers. They support this claim with the evidence that the answers under attack are 10% less natural compared to the original ones. My question is, why is a 10% difference considered a small one?
>
> During our human evaluation, we present the annotators with responses to the questions generated with and without the image being adversarially perturbed. On the image-unrelated subset, human rated 86% of the responses generated with the unperturbed image as natural, while 77% of the responses generated with the perturbed image are annotated as natural. Here, we concluded that in an absolute value, only about 10% sentences on average would make the user feel more unnatural. But what’s probably equally important is that 77% (37% for image-related) of the responses are natural to humans, therefore they wouldn’t realize something unintended has gone wrong. This is why we believe such an attack is stealthy.
>
> > If the current setting is realistic, I suggest showing the effectiveness of attacking the real-world system.
>
> Attacking an existing system is our current goal. We are working on blackbox transferability to be able to attack the well known systems with no publicly-available weights. But again, one of the goals of this work is to show new attack vectors and threats. Even though we were not able to carry our attack on real-world systems yet, the five attack objective we described in Table 1 are all following tool invocation syntax from real world systems (see Sec 3.1) to simulate the behavior as realistically as possible, which demonstrates the potential of our attack if they could be applied on real-world systems.

---

### Author Response · Authors · 2023-11-17

Dear Reviewers,

We revised the paper in response to the reviews and we highlighted the revisions in purple. In summary, our revision mainly clarifies that the tool invocation calls in Table 1 exactly follow the way that commercial LLMs are using, which suggests that our proposed attack scenario is highly realistic. We revised the background section for clarification and error correction. Additional experimental results were also included in the appendix.

---

### Meta-Review · Area_Chair_CJep · 2023-12-07

**Metareview:**

The paper shows that adversarial images can cause a multimodal language model to misuse tools.

Strengths:
- An important threat. Tool misuse is a much more important threat than the jailbreaks studied in many other works.

Weaknesses:
- Studies standalone models rather than a real system. These systems exist. They are not white-box of course, but this is likely what real attackers will have to deal with. So it would be nice to show this is possible.
- The result is somewhat unsurprising given that prior work has shown: (1) text-based attacks can cause tool misuse (e.g., Greshake et al); (2) adversarial images can cause LLMs to output adversarial text (e.g., Carlini et al). So the combination of the two of course works as well.

**Justification For Why Not Higher Score:**

Ultimately, reviewers all raised concerns about the lack of a study against a real system.
I tend to agree that this would be useful for bringing something new to this space, given that prior work has already demonstrated the vulnerabilities of white-box vision-language models.

**Justification For Why Not Lower Score:**

N/A

---

### Decision · Program_Chairs · 2024-01-16

Reject